# Equatorial waves as useful precursors to tropical cyclone occurrence and intensification

Xiangbo Feng [1] ✉, Gui-Ying Yang [1], Kevin I. Hodges[1] & John Methven [2]

Understanding and prediction of tropical cyclone (TC) activity on the medium range remains challenging. Here, we find that the pre-existing westward-moving equatorial waves can inform the risk of TC occurrence and intensification, based on a dataset obtained by synchronising objectively identified TCs and equatorial waves in a climate reanalysis. Globally, westward-moving equatorial waves can be precursors to 60–70% of pre-tropical cyclogenesis events, and to >80% of the events with the strongest vorticity, related to the favourable environmental conditions within the pouch of equatorial waves. We further find that when storms are in-phase with westward-moving equatorial waves, the intensification rate of TCs is augmented, whilst in other phases of the waves, storm intensity grows more slowly, or even decays. Coherent wave packets associated with TCs are identifiable up to two weeks ahead. Our findings show that westward-moving equatorial waves can be useful medium-range precursors to TC activity.

Tropical cyclones (TCs) are amongst the most destructive weather phenomena, causing intense rainfall, strong winds, and storm surges. Understanding and predicting TCs on medium-range timescales (3–15 days) and developing related early warning services could significantly reduce TC-related damage to economic activity and social welfare. Existing statistical and dynamical approaches have limited skill in predicting the medium-range risk of TCs[1–5]. This suggests that the key processes controlling the medium-range TC activity are not fully understood, and that numerical weather prediction (NWP) models have difficulty in simulating the underpinning processes. Identification of reliable precursors to TC activity, from days to a couple of weeks ahead, helps to understand and predict the medium-range risk of TCs.

Equatorial waves are important large-scale phenomena in the tropics[6], and have recently been associated with tropical cyclogenesis (TCG) in studies using the Best Track TC observations[7–9]. In these studies, the hypothesized mechanism is that equatorial waves are associated with enhanced convection in the background, favouring TC generation. In these studies, the TCG-wave relationship is basically determined from rainfall, which includes the TC rainfall, because the waves are extracted by applying a spectral filter to convection variables (i.e., precipitation and outgoing longwave radiation, OLR) in a space-time domain[10]. However, in addition to the convectively coupled equatorial waves, which have strong precipitation and OLR signals, there are many other equatorial waves, such as the equatorial waves identified by the horizontal velocity structures. The entire spectrum of equatorial waves may be connected to TC occurrence. In present NWP models, tropical rainfall variability is still poorly predicted on the medium-range timescales due to the lack of explicitly resolved convection[11–13]. This greatly reduces model skill in simulating the convectively coupled equatorial waves and associated TCG events[11–14]. In contrast, present NWP models have much better skill in predicting synoptic-scale wind anomaliess[15]. Hence, an approach for dynamical equatorial wave identification, which does not rely on convection, could leverage equatorial waves as predictable precursors to the risk of TCs. Another factor impeding our understanding of the simultaneous TCG-wave relationship is large uncertainty in TCG observations, due to the methods inconsistently used in TCG identification and for the storms obscured by clouds or monsoon troughs[16–18]. Furthermore, the time of emergence of the waves prior to TCG events and the impact of the waves on TC intensification have not been examined.

[1]National Centre for Atmospheric Science and Department of Meteorology, University of Reading, Reading, United Kingdom. [2]Department of Meteorology, University of Reading, Reading, United Kingdom. ✉e-mail: xiangbo.feng@reading.ac.uk

Here, we create a long-term (1980–2018) dataset that synchronizes TCs and dynamical equatorial waves based on the same global reanalysis using a phase-matching algorithm (see the Methods section). TCs and waves are consistently and objectively tracked from the European Centre for Medium-Range Weather Forecasts (ECMWF) fifth-generation reanalysis (ERA5)[19]. We derive equatorial waves by projecting dynamical fields (i.e., winds and geopotential height) at each pressure level onto theoretical equatorial wave modes[20,21], including westward-moving mixed Rossby-gravity (WMRG) waves and meridional mode number $n = 1$ and 2 Rossby (R1 and R2) waves. This approach is independent of convection, and not affected by the Doppler shift due to the background zonal flow. In this dataset, we trace the TC disturbance backwards in time before the declared time of TCG in the Best Track (hereafter termed "observed TCG event") to the earliest time that the disturbance can be identified by vorticity above threshold in ERA5[22,23]. On average, the pre-genesis disturbance can be identified ~4.6 days earlier than the observed TCG event. Thereafter, we will describe the vortices at this earlier stage as "pre-TC" features and the first identification in ERA5 as the "pre-TCG" event. Advantages of the pre-TCG event include a longer warning time in practice, a consistent identification process throughout the whole period, and a close association with the genesis-related environments[24]. Additionally, as the pre-TCG events are overall closer to the equator, they are expected to have a stronger relationship with equatorial waves.

Here, we find that the pre-existing westward-moving equatorial waves can modulate the pre-TCG events, and this effect becomes even stronger for the pre-TCG events with stronger cyclonic vorticity. For those events with the strongest cyclonic vorticity, globally, over 80% form in the favorable dynamical conditions associated with equatorial waves. The distinct feature of individual waves appears ~1 week before the time of the pre-TCG event, and the coherent wave packets emerge up to 10 days ahead. We find that the pre-existing dynamical waves can also significantly modulate storm intensification, with intensity growing faster in the favorable wave phase (i.e., two systems staying in-phase), followed by slower growth or even decay in the unfavorable phase (i.e., two systems staying out-of-phase). This study shows that the westward-moving dynamical equatorial waves are useful precursors to both TC occurrence and intensification. The diagnosed TC-wave relationship could potentially be used for predicting TC activity up to 2 weeks ahead.

## Results

### Relating equatorial waves to pre-TCG events

We first evaluate the occurrence of global pre-TCG events associated with the equatorial waves using their horizontal wind structures (Supplementary Table 1). In the Northern Hemisphere (NH), when the vortex of a pre-TCG event is in the cyclonic vorticity (positive relative vorticity) region of an equatorial wave, the two systems are defined as "matched in-phase", whilst they are "matched out-of-phase" when the storm vortex is in the anticyclonic (negative relative vorticity) region of the wave (see the Methods section for details). Equally, in the Southern Hemisphere (SH), a pre-TC vortex is in the negative relative vorticity region of an equatorial wave when they are described as "matched in-phase". We find that across the globe, 64% of 3459 pre-TCG events over 1980–2018 are matched in-phase to at least one type of westward-moving equatorial waves. The percentage contribution by each wave type in each ocean basin is provided in Supplementary Fig. 1. The Western North Pacific (WNP), Eastern North Pacific (ENP), and North Atlantic (NA) see the largest percentages (63–70%) associated with the westward-moving waves.

We find that the pre-TCG events with strong cyclonic vorticity are more often associated with westward-moving waves. On average, for those events (vortex >3 Cyclonic Vorticity Unit, CVU, 1 CVU = $10^{-5}$ s$^{-1}$), 73% of them are matched in-phase to at least one wave type (Supplementary Table 1). The percentage of events matched in-phased to

WMRG, R1, and R2 waves is 43%, 43%, and 39%, respectively. 38% of pre-TCG events are simultaneously matched in-phase to at least two types of the westward-moving waves, and 15% are matched to all three types of waves. Figure 1 shows the percentage contribution related to each wave type in each ocean basin for the pre-TCG events with the strong vortex (>3CVU). In the WNP, ENP, and NA, the overall percentage contribution is up to 74–88%, and each wave type is associated with ~50% of the events. The low percentages in the North Indian (NI), South Indian (SI), and South Pacific (SP) Oceans are associated with the relatively high proportion (~50%) of the pre-TCG events at which the vortex is moving eastward (Supplementary Fig. 2a), which are less likely affected by the fast westward-moving waves (more discussions in the next subsection).

We further divide the global pre-TCG events into sub-groups conditional on the cyclonic vorticity of the pre-TCG vortex. The contribution of pre-TCG events matched in-phase to the waves increases steadily from 58% for the events with the weakest vorticity to 100% with the strongest vorticity (Fig. 2). In contrast, the contribution of the pre-TCG events that are matched out-of-phase to the waves decreases from 58% for the events with the weakest vorticity to 40% with the strongest vorticity. When further conditioning on the type and combination of the waves, the percentage of in-phase pre-TCG events also increases significantly with the vorticity value of the pre-TCG vortex.

### Distinct features of equatorial waves related to pre-TCG events

The above relationship between the waves and pre-TCG events can be depicted by the wave structures composited on the pre-TCG vortex location. Figure 3 shows the longitude–time diagram of the composite of the wave meridional winds in the lower and upper troposphere, by leading and lagging pre-TCG events that are matched in-phase to the waves, in the NH. For these events, the pre-TCG vortex is located in a large-scale cyclonic area (with wavelength ~1500-3000 km) in the lower troposphere, with the southerly winds to the east and the northerly winds to the west, and in an anticyclonic area in the upper troposphere. The pre-TCG vortex is commonly matched to the cyclonic vorticity phase of WMRG, R1 and R2 waves. In contrast, in the NH, for those pre-TCG events that are matched out-of-phase to the waves (Supplementary Fig. 3), which occur much less often than the in-phase events, the pre-TCG vortex is commonly located in an anticyclonic area of the wave. We also calculate the composite of wind anomalies without applying the wave spectral filter (see the Methods section for details). Wind anomalies without projection onto the waves are well captured by dynamical equatorial waves (Supplementary Fig. 4), indicating the reliability of our approach in identifying the equatorial waves. We notice that in Fig. 3 and Supplementary Fig. 4, the area of significant composite of meridional winds is greater for the equatorial waves than for unfiltered wind anomalies, related to a higher level of noise in the latter field, indicating the benefit of using equatorial waves for TC precursors.

The wind fields composited relative to the vortex location and event time of pre-TCG exhibit intrinsic dynamical features of equatorial waves. First, the westward phase speed of the waves (6–10 m/s) is significantly faster than that of TCs (3–4 m/s) (Fig. 3), consistent with propagation of a wave structure independent of the movement of an established vortex. Previous study[25] using a similar wave identification method also showed that in the tropical Atlantic WMRG waves propagate westwards faster than TCs, while R1 and R2 waves propagate westwards more slowly in the lower troposphere. Secondly, the associated equatorial waves are identified up to 4 days ahead of the pre-TCG event in the lower troposphere and up to 7 days ahead in the upper troposphere. The signal of waves furthest from the pre-TCG vortex is 4000-6000 km to the east. Figure 4a–d further demonstrates the well detected low-level horizontal winds induced by the three types of simultaneously matched waves 0–3 days ahead of the pre-TCG event. Thirdly, for each wave type, when an in-phase pre-TCG event

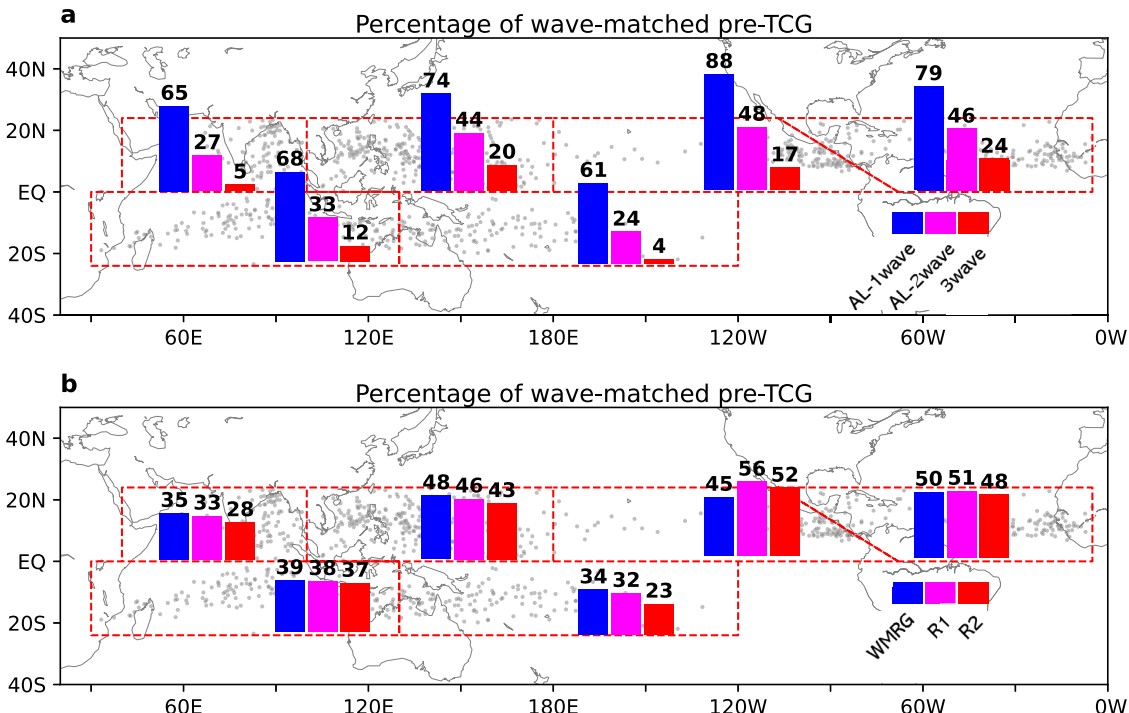

**Fig. 1 | Percentages of pre-tropical cyclogenesis (pre-TCG) events matched in-phase to equatorial waves. a** Percentages of pre-TCG events with strong vorticity matched in-phase to at least one type (AL-1wave, blue bar), at least two types (AL-2wave, magenta bar), and all three types of westward-moving waves (3wave, red bar), for each ocean basin, over 1980–2018. **b** as a, but for percentages of pre-TCG events with strong vorticity matched in-phase to each type of westward-moving waves: Westward-moving Mixed Rossby-Gravity wave (WMRG, blue bar), mode number 1 Rossby wave (R1, magenta bar), and mode number 2 Rossby wave (R2, red bar). Pre-TC intensity is defined by the absolute value of relative vorticity of the pre-TC vortex (in the units of Cyclonic Vorticity Unit, CVU, 1 CVU = $1.0 \times 10^{-5}$ s$^{-1}$); pre-TCG events with strong vorticity are the events when the pre-TCG vorticity >3CVU. Gray dots show the pre-TCG position in 24°N–24°S; red dashed lines show the boundaries of each ocean basin.

occurs, there is a coherent wave packet that propagates eastwards (Fig. 3). For example, in the NH and in the lower level, besides a cyclonic circulation around the pre-TCG vortex, there is a significant anticyclonic circulation on each side of this circulation cell (Fig. 4a). The multiple vorticity centers of the waves contrast with an isolated vortex of the pre-TC features. Furthermore, the eastward-propagating wave packets are statistically identifiable 7–10 days before the pre-TCG events. The group velocity of the waves varies, with WMRG waves having the largest group velocity (~5.5 m/s) among the three wave types. The early appearance of the waves is equally seen in the SH (Supplementary Figs. 5, 6). All these point to the fundamental distinction of the westward-moving waves associated with the pre-TCG events from the typical structure of the isolated TC vortex.

Next, we elucidate how the dynamical structure of the westward-moving waves could affect the pre-TCG events. The equatorial waves impose a cyclonic circulation around the pre-TC vortex at the event time of pre-TCG in the lower troposphere (Fig. 4a). We find that in the cyclonic vorticity area with closed circulation associated with equatorial waves (i.e., the wave "pouch"), middle-level relative humidity is anomalously higher than that in the surrounding areas, by ~5–10% for absolute value in the NH (Fig. 4e). At the event time, the maximum relative humidity occurs to the right of the pouch centre. These dynamical features (i.e., cyclonic vorticity and higher relative humidity) related to equatorial waves are identifiable a few days ahead in the east of the pre-TCG vortex (Fig. 4b–d, f–h). The propagation and evolution of circular area and moist air presented here conceptually resemble the "marsupial pouch" mechanism proposed for the TCG events resulting from easterly waves[26–28], though the wave types are different. Likewise, the pouch of equatorial waves could retain the enhanced moisture from dry air intrusion, providing a seeding bed for pre-genesis disturbance. Secondly, the low-level cyclonic condition of

equatorial waves is also associated with an anomalous convergent inflow at the lower level and divergent outflow at the upper level (Supplementary Fig. 7). The meridional convergent flow from the equatorward flank could further contribute to the increased relative humidity to the right of cyclonic circulation centre (Fig. 4e–h), even when the cyclonic circulation is not closed. The fluctuation of the environmental conditions related to the cyclonic vorticity phase of equatorial waves is also seen in the SH (Supplementary Figs. 6, 8). Thus, we anticipate that the favorable environmental conditions in the pouch of pre-existing westward-moving equatorial waves are likely related to the pre-TCG occurrence.

## Relating equatorial waves to pre-tropical cyclone intensification
We also evaluate the effect of the westward-moving equatorial waves on pre-TC intensification. In 7 days following the pre-TCG event, although some storms are developing into the TC features (as the pre-TC stage lasts 4.6 days on average), for simplicity, we still call them "pre-TCs" over these 7 days. Figure 5a shows the evolution of global pre-TC intensity in the 7 days after the pre-TCG events when initially matched in-phase to the westward-moving equatorial waves. The 24-hour intensification rate is shown in Supplementary Fig. 9. We find that the westward-moving waves, in which the pre-TCG vortex is initially embedded, can also significantly modulate the storm intensification afterwards. At the pre-TCG events, the storm vortices matched in-phase to the waves are stronger than those not matched. We also find that during the first two days after the pre-TCG event, the phase-matched pre-TCs intensify faster than other pre-TCs. The benefit gained from the waves starts to decline from day 2 as the fast-moving waves propagate ahead of the storm, leading to a large-scale environmental condition that suppresses pre-TC intensification (Fig. 3 and Supplementary Fig. 5). From day 4, the intensification rate remains

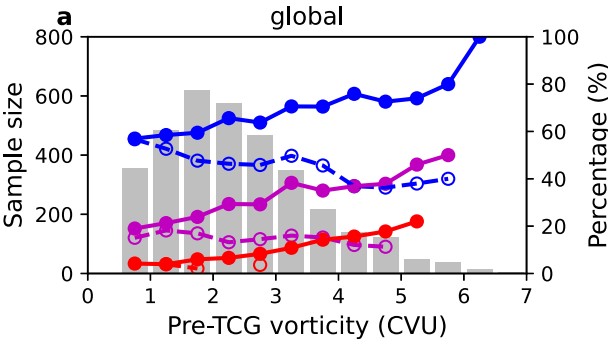

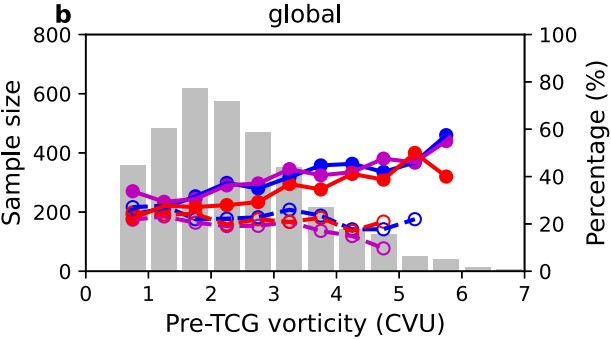

**Fig. 2 | Percentages of global pre-tropical cyclogenesis (pre-TCG) events related to equatorial waves conditional on storm intensity. a** Percentages of global pre-TCG events matched in-phase to at least one type (blue solid line with closed circle), at least two types (magenta solid line with closed circle), and three types of westward-moving waves (red solid line with closed circle), conditional on intensity of pre-TCG vortex, over 1980–2018. **a** Dashed lines are for percentages of pre-TCG events matched out-of-phase to at least one type (blue dashed line with open circle), at least two types (magenta dashed line with open circle), and three types of westward-moving waves (red dashed line with open circle). **b** as **a**, but for percentages of pre-TCG events matched in-phase to each type of westward-moving waves: Westward-moving Mixed Rossby-Gravity wave (WMRG, blue line with closed circle), mode number 1 Rossby wave (R1, magenta line with closed circle), and mode number 2 Rossby wave (R2, red line with closed circle). **b** Dashed lines show the percentages of pre-TCG events matched out-of-phase to each type of westward-moving wave: WMRG wave (blue dashed line with open circle), R1 wave (magenta dashed line with open circle), and R2 wave (red dashed line with open circle). **a**, **b** Gray bars show the sample size of pre-TCG events for each intensity group. The intensity of pre-TCG vortex is defined by the absolute value of relative vorticity of the pre-TCG vortex (in the units of Cyclonic Vorticity Unit, CVU, 1 CVU = $1.0 \times 10^{-5}$ s$^{-1}$).

small. We further confirm that the westward zonally propagating pre-TCs dominate the storm intensity-wave relationship, when the two systems have a longer time to stay in the vicinity of each other. In contrast, for the eastward-moving or recurving pre-TCs in the tropics, due to the fast separation of the two phenomena after the initial interaction, the waves have little effect on pre-TC intensity and intensification rate (Supplementary Fig. 10).

The phase-dependent effect of the waves on pre-TC intensification varies greatly with the basin (Fig. 5b–g and Supplementary Fig. 9b–g). The strongest effect is seen in the NI, SI, and ENP. The fast transition of dynamical fields related to the waves in the NI, compared to the slow zonal propagation of pre-TCs, is illustrated in Fig. 6a–d. The in-phase waves encourage pre-TC intensification in the first 2 days when the storms are still within the wave pouch. From day 4 to 6, the out-of-phase waves turn to suppress storm intensification, related to the large-scale anticyclonic circulation, low-level divergence, and upper-level convergence. We further find that the above effect of the waves is largely contributed by WMRG waves (Fig. 6e–h), related to a

faster westward-moving phase speed (Fig. 3). In the WNP, the waves can strongly affect the vortex intensity of the pre-TCG events but hardly modulate the storm intensification afterwards. Most of the WNP TCs form in the open ocean east of the Philippines and travel westwards to the Pacific Warm Pool. We confirm that the intensification of the pre-TCs formed east of the Philippines has no clear relationship with the wave phase (Supplementary Fig. 11). In contrast, for the pre-TCs formed inside the Warm Pool (i.e., the South China Sea), the phase-dependent effect of the waves on pre-TC intensification becomes clearer. In the NA, the waves barely affect pre-TC intensification either. In the SP, many pre-TCs move eastward (54% of TCs, Supplementary Fig. 2b) and separate quickly from the westward-moving waves, likely leading to an indistinguishable intensification-wave relationship.

## Discussion

Here, we find that the early appearance of westward-moving equatorial waves is associated with the generation of most pre-TCG events in the tropics, especially for the pre-TCG events with strong cyclonic vorticity. The pre-existing equatorial waves are associated with a larger proportion of eventual pre-TC generation in the WNP, NEP, and NA (75–90%), compared to in the NI, SI, and SP (60–65%). We also find that when the pre-TC vortices and waves are in-phase, pre-TC intensity tends to grow fast, while the following wave phase (out-of-phase) tends to inhibit pre-TC intensification. The effect on storm intensification varies greatly with the basin. We further show that the TC-wave relationship is likely associated with the modulation of environmental conditions by the pouch of equatorial waves.

We notice uncertainty in the above TC-wave relationship at regional and seasonal scales, related to characteristics of regional TCs and equatorial waves. A compound study conditional on equatorial waves and other factors, including long-term climate variability (e.g., ENSO) and large-scale circulations (e.g., monsoon trough), is desirable for a future investigation. Furthermore, the equatorial wave parameters used in this study (e.g., amplitude, wavelength, and phase speed) may vary with the wave identification methods and the wave band filter[10], and this could be another source of uncertainty in the TC-wave relationship. Additionally, we cannot entirely rule out the effect of TCs on equatorial wave identification, although the impact is expected to be small in the pre-TC stage. The relatively smaller-scale equatorial Rossby waves have a slower phase speed, and they could spend a long time interacting with TCs. This indicates that the two-way coupling between these waves and TCs might be more active and complicated than the coupling with other fast westward-moving waves. Evaluation of the uncertainty in both equatorial waves and their relationship with TCs will further benefit the understanding of the TC-wave interaction and using equatorial waves as the medium-range precursors for TC in practice.

The TC-wave relationship in this study can be converted to a medium-range probabilistic forecast for regional TC occurrence and intensification, conditional on the early appearance of equatorial waves. In this study, we used vorticity tracks identified to trace TCs back to an earlier stage before the declared TCG event in the Best Track data. The pre-TCG event can be identified 4–6 days before the observed TCG event, and the precursor wave packets can be identified 7–10 days before the pre-TCG event, indicating that a wave-dependent forecast for TC generation could have a longer warning time (about 14 days) in practice than waiting for genesis events to be declared in observations. A forecast for TC intensification with a similar warning time could also be setup based on the intensification-wave relationship. Furthermore, current global NWP models exhibit skill in predicting dynamical westward-moving equatorial waves[15], contrasting with a lower performance in TC prediction[1–5,29]. This suggests an opportunity to further develop the TC-wave relationship to a hybrid forecast of TCs, which can potentially extend the forecast lead time to a few weeks.

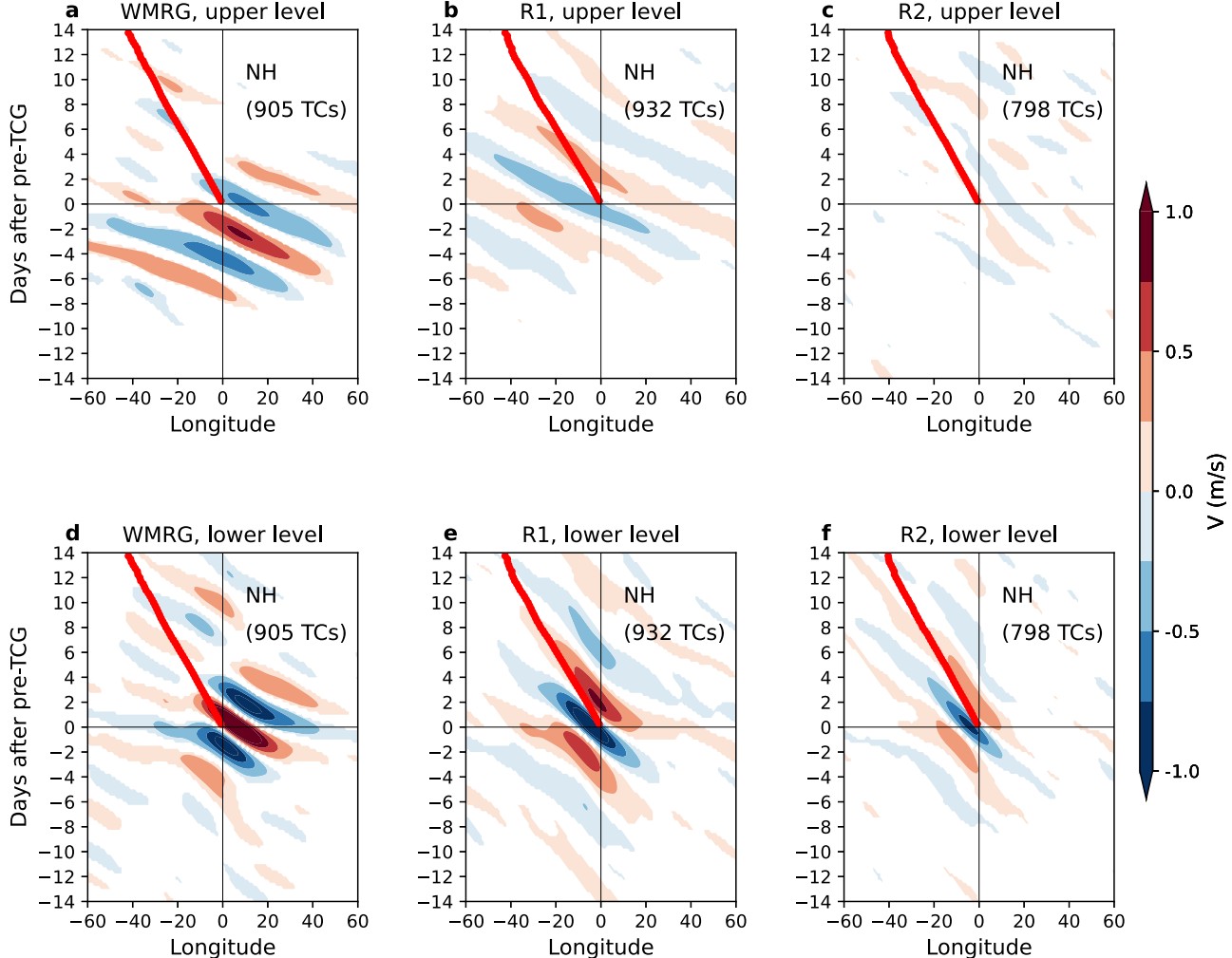

**Fig. 3 | Hovmöller diagram of equatorial waves onto pre-tropical cyclogenesis (pre-TCG) events in the Northern Hemisphere (NH). a** Hovmöller diagram of composite means (shading) of Westward-moving Mixed Rossby-Gravity wave (WMRG) meridional winds (*V*) at the equator (0°N) in the upper troposphere onto pre-TCG events that are matched in-phase to the waves, with respect to the vortex longitude and event time of pre-TCG, in the NH. Only composite significant at the 95% confidence level is shown. The total number of pre-TCG events matched in-phase to each type of wave is provided in the top right; thick red line shows the averaged longitude of storm track, with respect to the vortex longitude and event time of pre-TCG. **b**, **c** as **a**, but for mode number 1 Rossby wave (R1) and mode number 2 Rossby wave (R2) meridional winds at 8°N and 13°N, respectively. 0°N, 8°N, and 13°N are chosen because WMRG, R1 and R2 waves have the maximum values of *V* at these three latitudes, respectively. **d**–**f** as **a**–**c**, but for the lower troposphere. The upper troposphere is equally averaged over 100, 150, 200, 250, and 300 hPa, while the lower troposphere is averaged over 900, 850, 800, 750, and 700 hPa.

## Methods

### TC track data

The TRACK method[22,23], which was developed for cyclonic weather system tracking in climate reanalysis and weather forecasts[24,30], is used in our study to identity and trace cyclonic vorticity centers in six-hourly atmospheric data from the ECMWF fifth generation climate reanalysis (ERA5)[19]. The TRACK scheme used in this paper includes the following processes. First, the relative vorticity between 850 and 600 hPa is averaged on the T63 horizontal grid. A spatial filter with spherical harmonics is then applied to the averaged relative vorticity. The large-scale background is removed from the field by applying a high-pass filter ($n > 5$). After this, the positions of maximum vorticity centers in the Northern Hemisphere (or minimum vorticity in the SH) on the T63 grid are located. These positions are then used as the initial points to estimate off-grid locations using B-spline interpolation and maximization approaches. Then, during the data time series, when the value of vorticity exceeds the threshold of 0.5 Cyclonic Vorticity Unit (CVU, with 1.0 CVU = $1.0 \times 10^{-5}$ s$^{-1}$) in the range

0°–60°N (or the value is below the threshold of −0.5 CVU in the range 0°–60°S), these vorticity centers are identified. The tracking is produced by initializing a set of track points with a nearest neighbor algorithm and fitting them to a cost minimization function for track smoothness conditional on adaptive restrictions on track smoothness and displacing distance in a time interval. After the tracking is complete, a recursive search for the vorticity maxima within a 5° radius (geodesic) of the tracked centre at levels of 850, 700, 600, 500, 400, 300, and 200 hPa is carried out on the T63 grid, and these positions and vorticity values are added to the tracks. This can be utilized to examine whether a coherent vertical structure and a warm core are existing in the tracked storms.

In the TRACK method, the normal approach to TC identification in reanalyses is to apply criteria for intensity and the existence of a warm core (as above). However, this approach is different from observations and thus mismatches may occur between observed TCs and the objectively identified TCs in reanalyses[23]. For example, observed TCs are not identified at the same time and in the same region as in ERA5, or

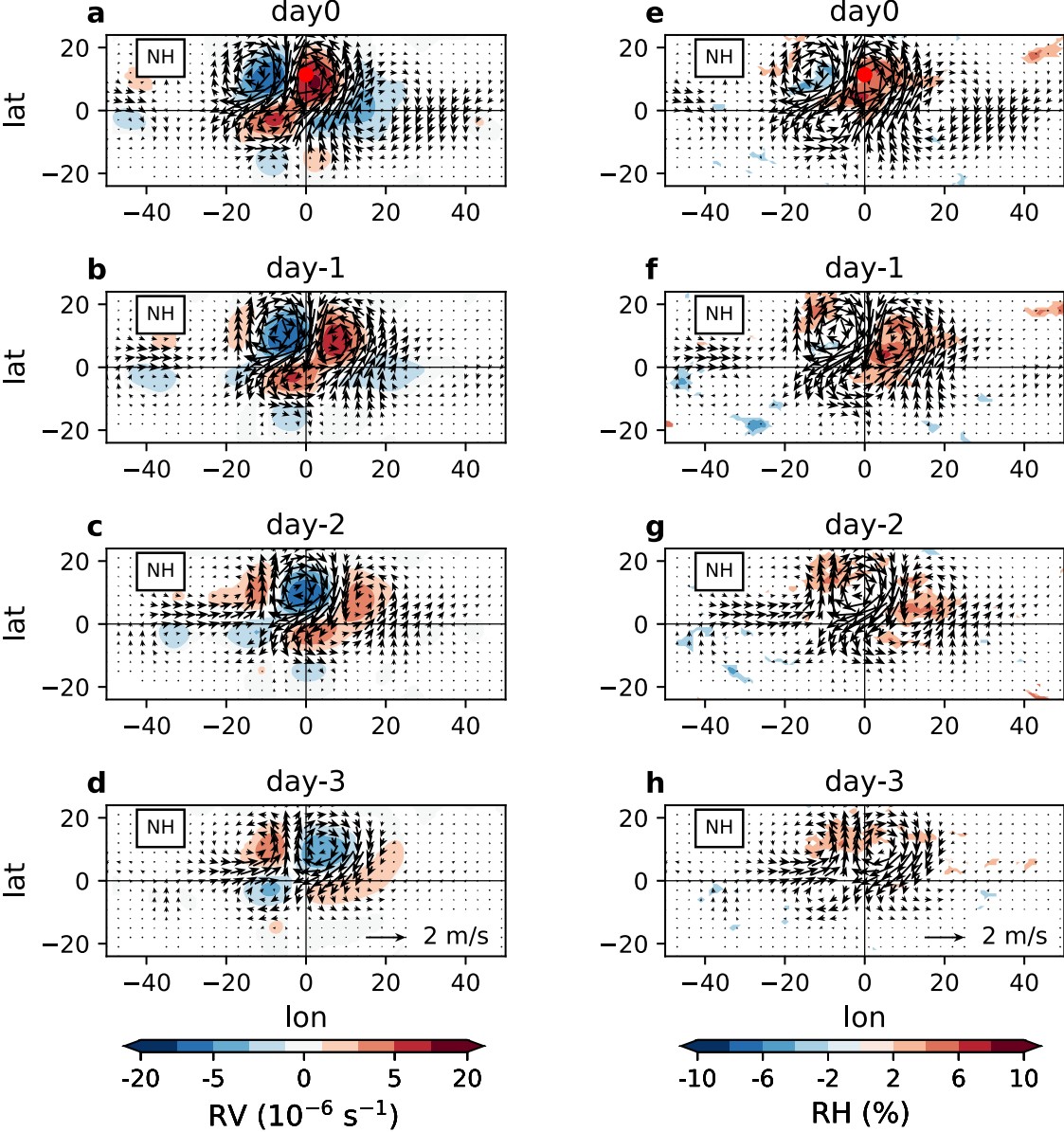

**Fig. 4 | Co-existing and pre-existing equatorial waves related to pre-tropical cyclogenesis (pre-TCG) events in the Northern Hemisphere (NH). a** Composite means of combined wave horizontal winds (vectors) and relative vorticity (RV, in the units of $10^{-6}$ s$^{-1}$; shading) in the lower troposphere onto pre-TCG events that are simultaneously matched in-phase to Westward-moving Mixed Rossby-Gravity wave (WMRG), mode number 1 Rossby wave (R1) and mode number 2 Rossby wave (R2), with respect to the vortex longitude and event time of pre-TCG, in the NH. Only composite of relative vorticity of wave winds significant at the 95% confidence level is shown. Winds are combined from WMRG, R1 and R2 wave winds. Red dot shows the averaged latitude of pre-TCG vortex. **b**–**d** as **a**, but for composite means 1–3 days before pre-TCG events. **e**–**h** as **a**–**d**, but with shading showing composite means of relative humidity anomaly at 700 hPa (RH, in the units of %). The lower troposphere is equally averaged over 900, 850, 800, 750, and 700 hPa.

ERA5 tracks are not observed in the Best Track. To avoid this inconsistency, a matching process is applied to match the ERA5 tracks against the observed Best Track from IBTrACS[31]. Precisely, an ERA5 track is matched to an IBTrACS track if the mean spatial separation is ≤5° over the corresponding paired track points and it is the track with the smallest separation. This process, without any TC objective identification, ensures that the final ERA5 tracks are those storms that are also observed in IBTrACS.

Note that in our study the ERA5 storms have an extended lifecycle consistent throughout the whole period (1980–2018), allowing the analysis of the "pre-TC" features, while IBTrACS has large inter-agency uncertainty in recording TCG events and pre-TC features[16–18]. ERA5 "pre-TCG" is the first point of the above identified TC track in ERA5, which is at an earlier stage than the genesis in IBTrACS (normally the first track point reaching the tropical storm intensity). To demonstrate this difference, we truncate the ERA5 tracks to the same length as the IBTrACS tracks (i.e., with the same lifetime). Supplementary Fig. 12 shows the differences in the time and intensity (i.e., relative vorticity at the vorticity centre) between ERA5 pre-TCG vortex, and ERA5 TC vortex at the observed TCG time, for the same TCs. On average, the ERA5 pre-TCG events are 4.6 days earlier in time, and 2.7 CVU weaker in relative vorticity of the vortex, than the TCG events at the observed TCG time.

For the ERA5 TC intensity, we use the absolute values of relative vorticity of the vortex (in the units of CVU) at 850 hPa. We also tested the results using other intensity variables (i.e., maximum 10 m wind speed and minimum sea level pressure) in the ERA5 tracks, and the conclusions are not changed. Observed TC intensity in IBTrACS was

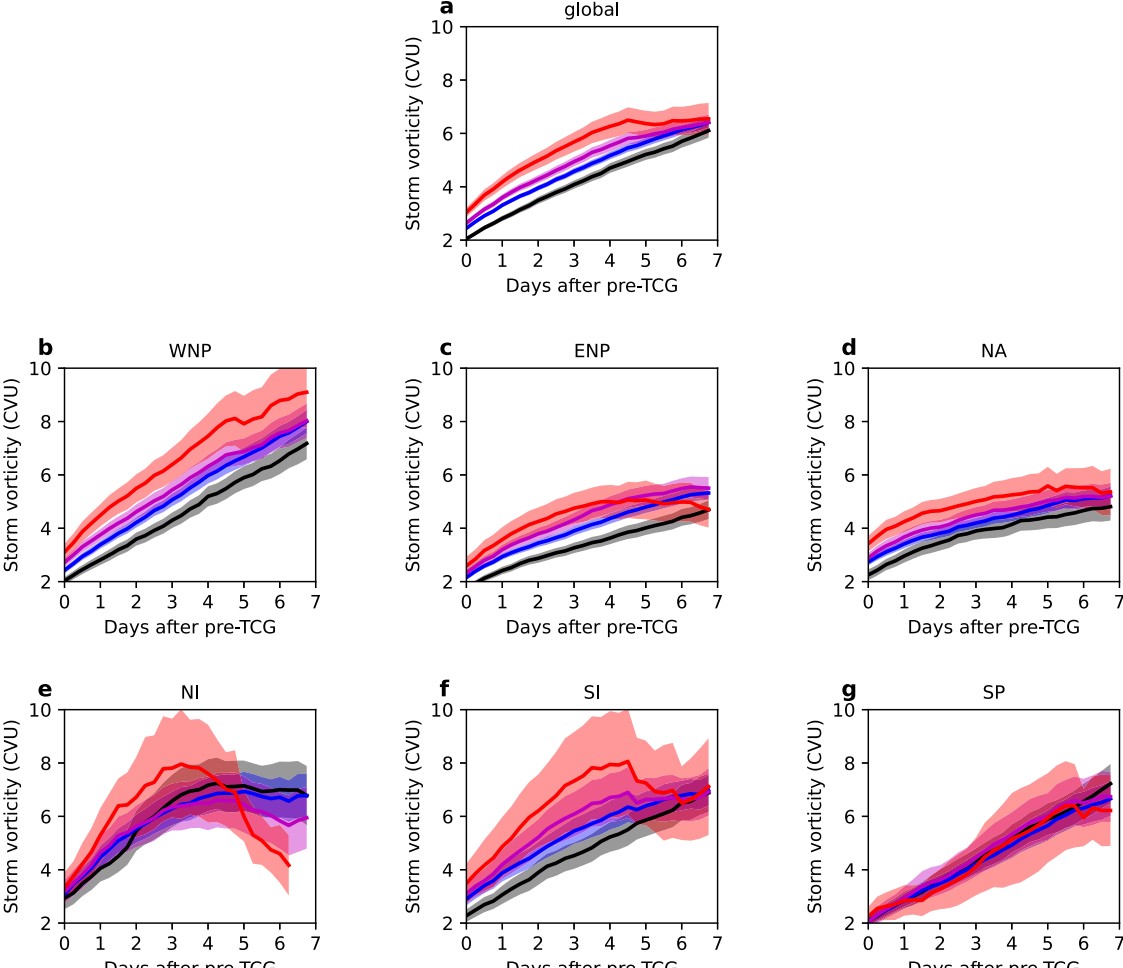

**Fig. 5 | Global and basin-wide pre-tropical cyclone (pre-TC) intensity related to equatorial waves. a** Global pre-TC intensity conditional on the phase matching in pre-tropical cyclogenesis (pre-TCG) events and equatorial waves, as a function of the time after pre-TCG events. The phase-matching includes: not matched in-phase to any type of westward-moving waves (black line), matched in phase to at least one type (blue line), matched in-phase to at least two types (magenta line), and matched in-phase to all three types of westward-moving waves (red line). The shading shows the 95% confidence interval of the mean. Pre-TC intensity is defined by the absolute value of relative vorticity of the pre-TC vortex (in the units of Cyclonic Vorticity Unit, CVU, 1 CVU = $1.0 \times 10^{-5}$ s$^{-1}$). The minimum sample size for each 6-hourly time interval is 10 storms. Westward-moving waves include Westward-moving Mixed Rossby-Gravity wave (WMRG), mode number 1 Rossby wave (R1) and mode number 2 Rossby wave (R2). **b**–**g** as **a**, but for pre-TCs in each ocean basin. WNP Western North Pacific, ENP Eastern North Pacific, NA North Atlantic, NI North Indian Ocean, SI South Indian Ocean, and SP South Pacific (ocean basins are defined in Fig. 1).

not used here because the intensity data are often missing especially for the early stages, including TCG, and because the observed tracks could mismatch the ERA5 equatorial waves. We notice that TC intensities are usually underestimated in climate reanalyses[23], meaning that the ERA5 TCs might be weaker than the observed. This might have an impact on the relationship with equatorial waves. However, the ERA5 tracks have merit in pre-TC features as they are objectively identified throughout the whole period.

We analyze the ERA5 TC tracks that originate in 24°S–24°N, to be consistent with the dynamical equatorial wave data. The TC tracks have a six-hourly interval, covering all seasons during 1980–2018. We also separate the ERA5 TC tracks into westward-moving and eastward-moving tracks[24]. For a pre-TCG event with westward-moving vortex, the 2nd track point must be further west than the 1st track point (at the pre-TCG time). Equally, for a pre-TCG event with eastward-moving vortex, the 2nd track point must be further east than the pre-TCG point. For the westward-moving TCs, they meet the following criterion: the poleward-most track point, or the track point closest to 24°N and 24°S, whenever it reaches first, is further west than the vortex centre of pre-TCG by at least 5°. The eastward-moving TCs are the storms that do not meet the above criterion.

Supplementary Fig. 2 shows the proportions of westward-moving pre-TCG events and TCs in each basin.

**Equatorial wave data**

In this study, dynamical equatorial waves are derived by projecting global wind and geopotential height data onto an orthogonal basis defined by the horizontal equatorial wave structures obtained from the theory of disturbances to a resting atmosphere on the equatorial $\beta$-plane[20]. Six-hourly horizontal winds and geopotential height in ERA5 are used here. This method identifies horizontal wind ($u$, $v$) and geopotential height ($Z$) structures associated with distinct equatorial waves. More precisely, potential equatorial waves are identified by projecting $u$, $v$, and $Z$ in the tropics (24°S–24°N) at each pressure level onto the different equatorial wave modes, using their sinusoidal structure in the zonal direction and parabolic cylinder functions in the meridional direction. These basis functions used for the wave projection are orthogonal, meaning that the wave structures here are orthogonal since they are pre-described as a series of the basis functions. In the parabolic cylinder functions, the meridional trapping scale is $y_0 = 6°$. The value of $y_0$ is pre-determined from the best fit of the theoretical equatorial wave solutions to observational data[20,21], and the

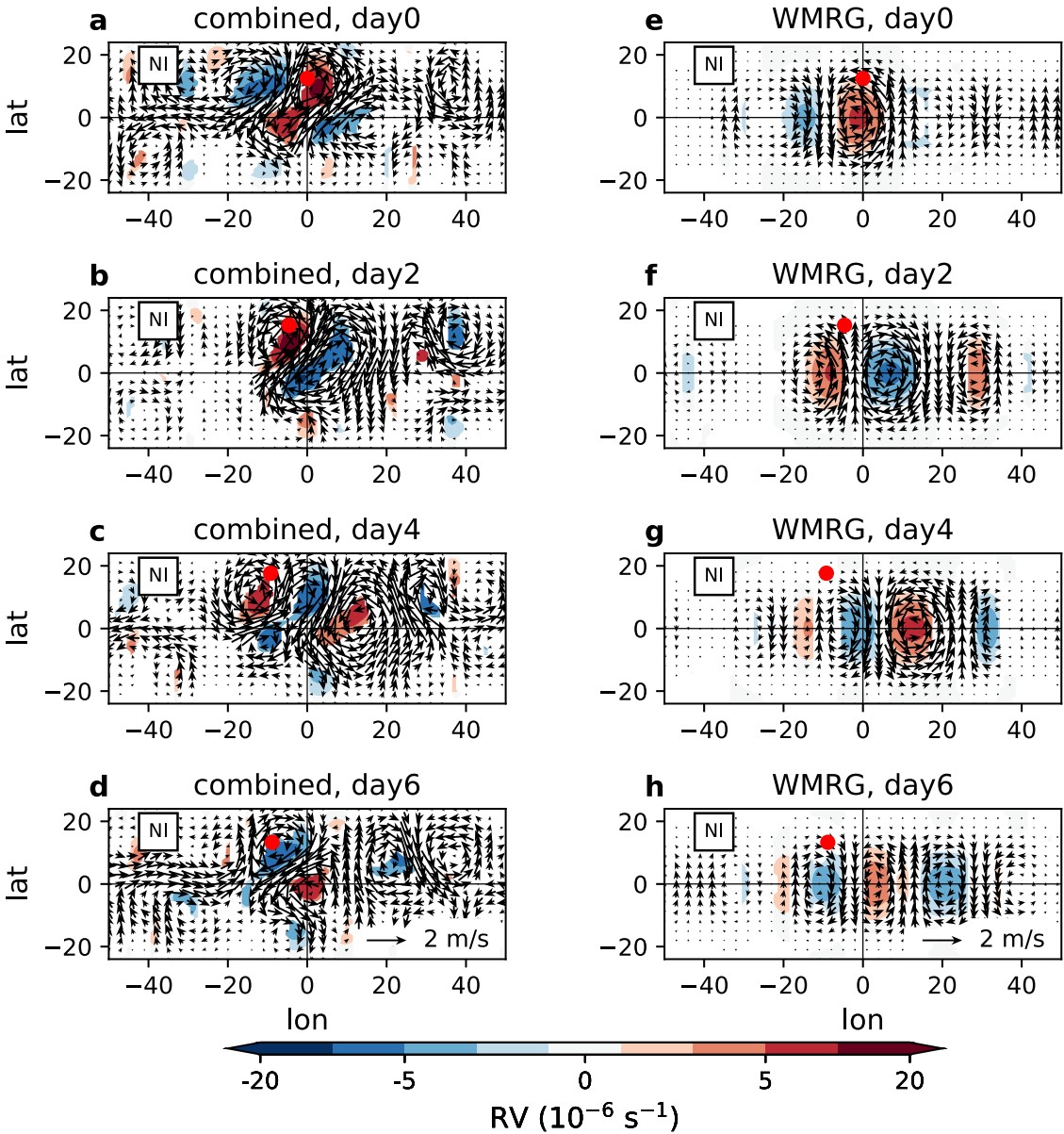

**Fig. 6 | Effect of equatorial waves on pre-tropical cyclone (pre-TC) intensity in the North Indian Ocean (NI). a** Composite means of combined wave horizontal winds (vectors) and relative vorticity (RV, in the units of $10^{-6}$ s$^{-1}$; shading) in the lower troposphere onto pre-tropical cyclogenesis (pre-TCG) events that are simultaneously matched in-phase to Westward-moving Mixed Rossby-Gravity wave (WMRG), mode number 1 Rossby wave (R1) and mode number 2 Rossby wave (R2), with respect to the vortex longitude and event time of pre-TCG, in the NI. Only composite of relative vorticity of wave winds significant at the 95% confidence level is shown. Winds are combined from WMRG, R1, and R2 wave winds. Red dot shows the averaged latitude of pre-TCG vortex. **b–d** as **a**, but for composite means 2–6 days after pre-TCG events. **e–h** as **a–d**, but for composite means of WMRG wave horizontal winds and relative vorticity onto pre-TCG events that are matched in-phase to WMRG waves. The lower troposphere is equally averaged over 900, 850, 800, 750, and 700 hPa.

structures of equatorial waves are not in fact sensitive to the choice of $y_0$. Before the projection, a broad-band spectral filter, with wavenumber 3–40 and period 2–10 days, is applied to separate eastward and westward-moving waves. The broad-band filter can account for Doppler shift of frequencies by the background zonal flow. Please note this wave identification method does not directly apply the linear adiabatic theory for equatorial waves in a resting atmosphere. In particular, the dispersion relation and vertical structure are not imposed, because in reality, these aspects are sensitive to any background zonal flow that varies with height and time. This method allows wave properties to emerge from the dynamical data.

Our ERA5 wave dataset contains three equatorial wave modes: westward-moving mixed Rossby-gravity (WMRG) and meridional mode number n = 1 and 2 Rossby (R1 and R2) waves. In our study, we focus on the westward-moving equatorial waves because they travel in the same direction as most of TCs. The wave dataset spans 39 years from 1980 to 2018 covering all seasons, with a 6-hourly interval at 1° resolution on 28 pressure levels from 1000 to 70 hPa. We note that since the meridional winds of WMRG, R1, and R2 waves represent the key structures of these waves, wave activities can be represented by the variables at a given latitude. A latitude, depending on the wave type, is selected to capture the peak of wind velocity of waves: 0°N for WMRG waves, 8°N for R1 waves and 13°N for R2 waves in the NH, and 8°S for R1 waves and 13°S for R2 waves in the SH.

## Matching pre-TCG and equatorial waves
There are two main steps in linking the pre-TCG events to equatorial waves.

Step1: we diagnose the features of dynamical equatorial waves around the pre-TCG location. Supplementary Fig. 13 shows the composited winds of westward-moving equatorial waves at each pressure level onto the vortex centre of all pre-TCG events in the globe (24°S–24°N) over 1980–2018. Westward-moving equatorial waves show a very coherent baroclinic vertical structure around the vortex centre of pre-TCG, e.g., with southerly winds to the east and northerly winds to the west in the lower troposphere, and with reversed winds in the upper troposphere, in the NH. These features are opposite in the SH. This means that the longitude of the pre-TCG vortex is generally located in the cyclonic vorticity area of the equatorial waves, which guides our next step to design the following phase-matching process for the pre-TCG events and the waves.

Step2: the phase and amplitude of the composited waves at the low level (equally average over 900, 850, 800, 750, 700 hPa) are used to define the criteria in the phase-matching process. The phase-matching is to decide the phase relationship between the pre-TCG events and the coincident equatorial waves. In the NH, the matching process is described as:

- For the WMRG wave, a pre-TCG event is considered to be "matched in-phase" when (i) the wave meridional wind averaged east of the pre-TCG vortex centre at the equator (3–13° longitude relative to the pre-TCG vortex centre, at 0°N; green bar in Supplementary Fig. 14a) is southerly ($V > 0$), and (ii) the amplitude of this average wind is more than the half standard deviation of $V$ in the same longitude band ($V > 0.5 \times V_{STD}$). Equally, they are considered "out-of-phase" when the averaged wind is northerly ($V < 0$) and $V < -0.5 \times V_{STD}$.
- For the R1 or R2 waves, a pre-TCG event is considered to be "matched in-phase" when (i) the wave wind averaged west of the pre-TCG vortex centre at 8°N or 13°N (−10–0° longitude relative to the pre-TCG vortex centre; green bar in Supplementary Fig. 14b, c) is northerly ($V < 0$), and (ii) the averaged wind amplitude is more than the half standard deviation of $V$ over the same band ($V < -0.5 \times V_{STD}$). Equally, they are considered "out-of-phase" when the averaged wind is southerly ($V > 0$) and $V > 0.5 \times V_{STD}$.

The wind is averaged at 0°N for the WMRG wave, 8°N for the R1 wave, and 13°N for the R2 has because, as mentioned above, the waves have the maximum values of $V$ at these three latitudes, respectively. The same process is used for the pre-TCG events in the SH, but with the opposite wind criteria used for WMRG (for "in-phase matching", $V < -0.5 \times V_{STD}$ at 0°N), R1 (for "in-phase matching", $V > 0.5 \times V_{STD}$ at 8°S) and R2 (for "in-phase matching", $V > 0.5 \times V_{STD}$ at 13°S) waves.

We tested the sensitivity of our results to the matching thresholds, by, e.g., changing the threshold values from 0.5 times standard deviation of wind speed, to 0.75 and 1.0 times. We found that although increasing (or decreasing) the thresholds can decrease (or increase) the sample size of the matched pre-TCG events and slightly alter the longevity of wave appearance, the conclusions remained unchanged. The positions of green bars in Supplementary Fig. 14 are largely determined by the composite of equatorial waves onto all pre-TCG events (Supplementary Fig. 13). The green bars are approximately half wavelength long. We tested various length and position of the green bars and found that the present expression provides the most robust results for global pre-TCG events.

The principle of this phase-matching algorithm is to ensure that the pre-TCG vortex sits in a circular environment induced by the equatorial waves. Here, this is determined by the two factors, which are (i) the relative position of the waves and the pre-TCG vortex centre, and (ii) the amplitude and direction of the low-level winds of the waves. We choose the low-level winds over the middle-level or upper-level winds in the matching algorithm because the structure of

the waves is compact and better organized in the lower troposphere (Supplementary Fig. 13). We choose the wind velocity over other variables because dynamical equatorial waves are directly depicted by the winds. Other wind-related estimates, such as relative vorticity, were also tested when we defined the matching algorithm, and they were found to produce similar results.

## Environmental conditions related to tropical cyclones

In this study, we also evaluate large-scale environmental conditions related to TC activity given the phase matching with the waves, based on the ERA5 data. These conditions include six-hourly relative vorticity and convergence, at the lower level (equally average over 900, 850, 800, 750, 700 hPa) and the upper level (equally average over 100, 150, 200, 250, 300 hPa), based on the already composited winds of the equatorial waves. Six-hourly anomalies of winds at the lower and upper levels, and relative humidity at 700 hPa, without applying the wave band filter, are also composited to the vortex centre and event time of pre-TCG. For these unfiltered anomalies, the 6-hourly climatology over 1980–2018 is first removed before the composite.

## Significance test in composite analysis

In the composite analysis of the waves and other related fields (e.g., relative vorticity, convergence, and relative humidity), the statistical significance is tested using the two-tailed Student's $t$ test with a $p$ value of 0.05. The tested null hypothesis is that the composite average is zero. The composite average is statistically significant if the average value is above or below zero at the 95% confidence level. The test is applied to the variables at every grid point and time interval (6 hourly). In the average analysis of storm intensity, the 95% confidence interval of the mean is calculated by assuming a standard normal distribution as in the two-tailed Student's $t$ test.

## Data availability

The ERA5 tropical cyclones and equatorial wave data generated in this study have been deposited in the Zenodo database under https://doi.org/10.5281/zenodo.7490606 (https://zenodo.org/record/7490606#.Y7Hyzi2l2xY). The data are available with open access. The ERA5 data used to identify tropical cyclones and equatorial waves are generated by ECMWF and distributed by the C3S CDS (https://cds.climate.copernicus.eu/#!/search?text=ERA5&type=dataset). IBTrACS (International Best Track Archive for Climate Stewardship Project, Version 4) tropical cyclones data are archived from NOAA National Centers for Environmental Information with https://doi.org/10.25921/82ty-9e16 (https://www.ncei.noaa.gov/access/metadata/landing-page/bin/iso?id=gov.noaa.ncdc:C01552). Computing and data storage facilities were provided by JASMIN (https://jasmin.ac.uk).

## Code availability

The code for tropical cyclone identification is available from https://gitlab.act.reading.ac.uk/track. The codes for equatorial wave identification and TC-wave matching are available from the corresponding author on request.

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

## Acknowledgements

X.F., G.Y.Y., K.I.H., and J.M. were supported by the UK Met Office Weather and Climate Science for Service Partnership for Southeast Asia (H5480800, H5525000), as part of the Newton Fund. X.F., K.I.H., and J.M. were also supported by the WWRP Tropical Cyclone-Probabilistic Forecast Products (TCPFP) project (Letter of Agreement, 25978/2021-1.6SI) coordinated by the World Meteorological Organization (WMO). G.Y.Y. was also supported by the National Centre for Atmospheric Science through the NERC National Capability International Programmes Award (NE/X006263/1).

## Author contributions

X.F. designed and performed the analysis; K.I.H. tracked the tropical cyclone tracks in the climate reanalysis (ERA5), and matched them to IBTrACS tracks; G.Y.Y. identified the equatorial waves in the climate reanalysis (ERA5); X.F., G.Y.Y., K.I.H., and J.M. discussed the results and wrote the manuscript.

## Competing interests

The authors declare no competing interests.
