## [Peer Review File · Nature Communications]

Equatorial waves as useful precursors to tropical cyclone occurrence and intensificationREVIEWER COMMENTS

Reviewer #1 (Remarks to the Author):

The manuscript investigates the role of equatorial waves (EW) as large scale precursors of tropical cyclones (TC) occurrence and intensification, which is a problem of practical relevance given its potential for improving medium range predictions of impactful weather events. The observational analysis indicates that westward moving equatorial waves modulate both TC frequency and intensity. This is a noteworthy result that contributes and expands the existing literature in at least two main areas. First, most of the previous studies are focused on the relationship between EW that are coupled to convection and TC whereas, here, the analysis is broadened to all EW that may or may not be convectively coupled. One interesting result is the relative weak influence of Kelvin waves compared to what was found for convectively coupled Kelvin waves - which could be a statement of the different mechanisms at play when considering coupled and dry EWs. Second, most studies that I am aware of, relate EW to frequency of occurrence of TC, meanwhile, the present study suggests that specific EW phases modulate TC intensity. That brings further information that might be useful for extended predictions of TC evolution.

While the results are certainly interesting and bring new food for thought, I have some general comments related to the motivation and methodology. And then some more specific major concerns about the methodology (see below). I think those can all be addressed on a round of major revisions.

Comment 1 -Motivation:

The motivation to revisit the relationship between EW and TC is carried out with an overall negative tone towards previous analysis that I find distracting and unnecessary for a few reasons. First, the work presented here nicely builds and expands on the previous literature more focused on the link between convectively coupled EW and TC. In other words, there is no need for comparative statements given that the goal in the present paper is to investigate the role of EW in modulating TC, without constraining the analysis to only EWs that are coupled to convection. Second, the overall methodology is largely inspired by those same previous studies, so once again, this is an extension, or a build up from what we have learned from the previous literature. The paper will get more attention from the community if the focus is on what we can learn from your approach, as opposed to what is wrong with previous approaches. Last, given the complexities involved in identifying EW, tracking TCs as well as their potential interactions, I feel like there is a need for multiple approaches, which might be a more positive way to motivate the work presented here.

Comment 2 - Methodology: This is partially related to my previous comment. While I entirely agree with the issue of contamination of TCs when filtering for EW using OLR or precipitation data, the method applied here has some of the same limitations. That is, because TC are compact structures, their circulation patterns project onto all modes of EWs, therefore TC information is potentially leaked to the modes of EW analyzed here. The changes in TC intensity with EWs phases, for example, could be an artifact of the projection method. Another potential issue is the projection of Easterly Waves onto the horizontal structures used for the projection. Because Easterly waves are known precursors for EW, a discussion on this issue seems appropriate. All methods are limited in some way, and it is important to bring up these caveats to readers' attention. It might inspire other researchers to develop new approaches that might circumvent some of these issues, for example.

Specific Comments:

L1: I don't think it makes a lot of sense to say "Dynamical equatorial waves" because all waves are dynamical, by definition. I think that the authors are trying to make a distinction between "coupled" and "dry" EW, but I think, for the most part, they could just say equatorial waves.

L52-55: This statement needs some clarification/re-wording. If convectively coupled EW were the main precursor for TC, then I agree with your statement. But if "dry" EW are important, then the inadequate simulation of convection should not be any barrier for leveraging EW for extended predictions of TC events.

L122-123: Have you looked at longitude-time diagrams of wind anomalies (without the EW filtering) to see if there is evidence of these westward moving disturbances without filtering for them? Also, a similar composite for TCs that are not associated with the EW identified here could offer a nice contrast to the cases where a EW precursor is identified.

L355 - 361: While the dispersion and vertical structure are not explicitly imposed, they are implicitly constrained through the choice of trapping scale for parabolic cylinder function. For example, the latitudes chosen to investigate WMRG, R1 and R2 are based on the choice of equatorial radius, which itself is related to the vertical structure of the wave, that also imposes the gravity wave speed. By the way, I don't think the value used for the trapping scale is mentioned anywhere, but maybe I missed it.

L363: If the projection is applied independently to each variable, does that mean that the WMRG, R1 and R2 are not orthonormal? Either way, a statement clarifying this point might be useful here.

Signed: Juliana Dias

Reviewer #2 (Remarks to the Author):

The authors hypothesize that equatorial waves (EWs) enhance the background convection, favouring tropical cyclone (TC) generation. To find out how the two features are related, they combine two methods for identifying EWs and TCs in ERA5 data in 1980-2018.

The EWs are analyzed using the method of Yang et al. that projects winds and geopotential fields on spatial structures of equatorial waves from adiabatic shallow-water equations on the equatorial beta plane. The method is one of methods discussed in a recent review paper by Knippertz et al. The three westward-propagating waves (R1 and R2 Rossby waves and the WMRG wave) are compared in time and space with the outputs of the TC tracking algorithm. Depending on the relative location of vorticity maxima and winds associated with the three waves, the events are considered "matched in-phase" or "out-of-phase".

Based on the collocation of EWs and TCs, the authors conclude that westward-moving EWs are responsible for 60-70% of TC genesis events and that EWs have the strongest effect on TC genesis in the North Pacific and North Atlantic. They also argue that EW signals are identifiable up to two weeks ahead and could be used as reliable precursors to TC activity, indicating an unprecedented potential for improving medium-range prediction of TCs.

The large EW variance found by the authors and the EW longevity are questionable given figures 16-17 in Knippertz et al. and their discussion of EW filtering. The matching algorithm requires more detailed analysis since the three EW types are largely defined by vorticity. The main issue with the manuscript is however, that the mechanism by which the two processes, EWs and TC are coupled, is left completely undiscussed. It is unclear how EWs are "responsible for TC genesis events", and how this analysis indicates "an unprecedented potential for improving medium-range prediction of TCs". Given several critical studies of EW forecasting cited in the paper, and availability of ERA5 forecasts, it should be possible to extend the analysis to forecast data and to look into the physical mechanism of how the two features may be coupled.

Reviewer #3 (Remarks to the Author):

The manuscript reports on the connection between westward-moving dynamical equatorial waves and global tropical cyclone (TC) genesis and intensification, identified using 39 years of ERA5 reanalysis data. The most noteworthy result is that these waves (WMRG, R1, and R2) are all found to be useful precursors for the occurrence and intensification of TCs. Other interesting findings are the heightened relationship for stronger genesis events, and the variations by region. Composites of horizontal wind fields yield further insights into the role of these equatorial waves during and shortly after the time of genesis.

The work is original, of significance to the field, and it maintains a high standard. The approach here of mapping the dynamical fields in the reanalysis data onto the theoretical westward-moving wave solutions is novel. It mostly complements the literature, which generally selects equatorial waves via their convective characteristics (OLR etc.). There are a few contradictions with the state of the field, indicated below. Overall, the work supports the conclusions and claims for WMRG, R1, and R2. The methodology mostly seems sound, with appropriate statistical and sensitivity tests to confirm the robustness of the results. The figures are clear. The manuscript is written transparently for the results to be recreated by an interested reader.

Overall, my recommendation is for acceptance subject to the following revisions:

Major Comments:

1. It is reported in the manuscript that "We also evaluated the effect of eastward-moving Kelvin waves on TC activity, and found that Kelvin waves do not have a clear impact on either TC genesis or TC intensification, associated with the fast separation of the TCs and Kelvin waves." In contrast, papers such as Schreck (2015) demonstrate the linkage between Convectively Coupled Kelvin Waves (CCKWs) and tropical cyclogenesis. Lawton et al. (2022) recently expanded on this, using composites based on 39 years of ERA5 data, to demonstrate the role of CCKWs on African Easterly Wave behavior. These and other papers cited therein are forming a consensus on this relationship. In the manuscript, are the Kelvin Waves convectively coupled, or are they more idealized "dry" Kelvin Waves? To support their claim above, the authors ought to summarize their methodology for finding the Kelvin Waves, and show the results (in the supplement, at least). Otherwise, I recommend that the authors remove their statements on Kelvin Waves if no evidence is provided to back them up.
2. The declaration of "TCG" in ERA5 employed here is inconsistent with how the TC field thinks about genesis. It is certainly beneficial here to include pre-TC disturbances in the ERA5 data, rather than beginning with the Best Track values of position and intensity. However, the time difference (average of 4.6 days) between the observed (IBTrACS) genesis and the "ERA5 TCG" is concerning. This is properly acknowledged in line 314, where the authors state that "ERA5 TCG is the first point of the identified TC track, which is an earlier stage than the genesis in IBTrACS". This suggests that ERA5 TCG should not be called TCG, and it is not yet part of a "TC track" as a TC is a few days away from forming. A reader might be misled into interpreting the "ERA5 TCG" location as the actual location of the onset of genesis, despite it being an average of 4.6 days, and sometimes even up to 20 days (according to Supplemental Figure 8) prior to the IBTrACS genesis time. I do see great value in the TC genesis process being utilized. It just does not correspond to the time of actual genesis. How about replacing "ERA5 TCG" with "Pre-Genesis Disturbance" or similar? The terminology is more cumbersome, but more accurate. If this is done, the methodology and results can remain as is, although the explanations will need to be refined to emphasize the processes prior to genesis.
3. An alternative to Major Comment 2, if the authors feel a strong need to do the phase matching at the genesis point, is to select the genesis location and time that are closest to those in IBTrACS. This

would be considerably more work, and perhaps less insightful than the current method which does not use the genesis point for the phase matching.

Minor Comments

1. Line 16. What is the "new observational dataset"? This study just uses ERA5, which is no longer new (but still state-of-the-art), and it is not "observational".

2. Lines 25 and 83. While this is an interesting study, it is overstated in places without evidence behind the statements. For example, the last sentence in the Abstract suggests an "unprecedented" potential for improving medium-range predictions of TCs, which seems too simplistic. I suggest removing this sentence, until the potential to add value to medium-range prediction of TCs has been demonstrated.

3. Line 43. It seems strange to say that "the identification of equatorial waves is contaminated by the heavy TC-related rainfall". Complicated, perhaps, but not contaminated. And it is not necessarily restricted to the TC-related rainfall.

4. Lines 62 and 63. The TCs are not observed. They are diagnosed in ERA5, as is described correctly in the paragraph beginning on line 302.

5. While the manuscript is largely well-written and straightforward to follow, there are several minor grammatical and typographical errors.

References:

Lawton, Q. A., Majumdar, S. J., Dotterer, K., Thorncroft, C., & Schreck, C. J., III., 2022. The Influence of Convectively Coupled Kelvin Waves on African Easterly Waves in a Wave-Following Framework, *Mon. Wea. Rev.*, 150, 2055-2072.

Schreck, C. J., 2015: Kelvin waves and tropical cyclogenesis: A global survey. *Mon. Wea. Rev.*, 143, 3996-4011.

Responses for Nature Communications: Paper # NCOMMS-22-31072-T

Title: Dynamical equatorial waves: precursors to tropical cyclone occurrence and intensification

We would like to thank the reviewers and editor for their careful reading and constructive comments on the previous version of the manuscript. We have worked to address all of the reviewer's comments carefully in this revised version. Our responses to the reviewers' comments are highlighted by blue text in the following.

Reviewer #1 (Remarks to the Author):

The manuscript investigates the role of equatorial waves (EW) as large scale precursors of tropical cyclones (TC) occurrence and intensification, which is a problem of practical relevance given its potential for improving medium range predictions of impactful weather events. The observational analysis indicates that westward moving equatorial waves modulate both TC frequency and intensity. This is a noteworthy result that contributes and expands the existing literature in at least two main areas. First, most of the previous studies are focused on the relationship between EW that are coupled to convection and TC whereas, here, the analysis is broadened to all EW that may or may not be convectively coupled. One interesting result is the relative weak influence of Kelvin waves compared to what was found for convectively coupled Kelvin waves - which could be a statement of the different mechanisms at play when considering coupled and dry EWs. Second, most studies that I am aware of, relate EW to frequency of occurrence of TC, meanwhile, the present study suggests that specific EW phases modulate TC intensity. That brings further information that might be useful for extended predictions of TC evolution.

While the results are certainly interesting and bring new food for thought, I have some general comments related to the motivation and methodology. And then some more specific major concerns about the methodology (see below). I think those can all be addressed on a round of major revisions.

We appreciate that the reviewer recognizes the research value of our paper. The reviewer's comments have been carefully considered and addressed as follows.

Comment 1 -Motivation:

The motivation to revisit the relationship between EW and TC is carried out with an overall negative tone towards previous analysis that I find distracting and unnecessary for a few reasons. First, the work presented here nicely builds and expands on the previous literature more focused on the link between convectively coupled EW and TC. In other words, there is no need for comparative statements given that the goal in the present paper is to investigate the role of EW in modulating TC, without constraining the analysis to only EWs that are coupled to convection. Second, the overall methodology is largely inspired by those same previous studies, so once again, this is an extension, or a build up from what we have learned from the previous literature. The paper will get more attention from the community if the focus is on what we can learn from your approach, as opposed to what is wrong with previous approaches. Last, given the complexities involved in identifying EW, tracking TCs as well as their potential interactions, I

feel like there is a need for multiple approaches, which might be a more positive way to motivate the work presented here.

We thank the reviewer for pointing this out. We agree that the present study was inspired by previous studies on TC-wave relationship, and we also agree that it is necessary to acknowledge the added value of having multiple approaches to identify the waves, including the associations with TCs. In the revised version, we have made changes in the 2nd paragraph of the Introduction section to reflect the improvement in the motivation.

We also removed the statement about other wave methods in the Methods section (the 1st paragraph in the “Equatorial wave data” subsection in the previous version), to tone down the caveats of other methods.

We hope these changes will address the reviewer’s concern.

Comment 2 - Methodology: This is partially related to my previous comment. While I entirely agree with the issue of contamination of TCs when filtering for EW using OLR or precipitation data, the method applied here has some of the same limitations. That is, because TC are compact structures, their circulation patterns project onto all modes of EWs, therefore TC information is potentially leaked to the modes of EW analyzed here. The changes in TC intensity with EWs phases, for example, could be an artifact of the projection method. Another potential issue is the projection of Easterly Waves onto the horizontal structures used for the projection. Because Easterly waves are known precursors for EW, a discussion on this issue seems appropriate. All methods are limited in some way, and it is important to bring up these caveats to readers' attention. It might inspire other researchers to develop new approaches that might circumvent some of these issues, for example.

We thank the reviewer for this valuable comment. We agree that the limitations of our method and the associated uncertainties should be explicitly discussed. We have improved our manuscript by addressing the following points:

- Regarding the contamination of EW identification by TCs. We agree that the horizontal wind and geopotential structures associated with TCs will project onto the spatial wave basis functions across a range of zonal wavenumbers and frequencies. However, the distinct dynamical features of the theoretical EWs including intrinsic phase speed and group velocity are observed in the composites constructed from the data. These properties are not built into the wave identification method or the broad band filter and therefore the existence of wave packets with distinct group velocity is evidence of equatorial wave behaviour that is independent of TC dynamics. We have a paragraph in the manuscript to discuss the intrinsic characteristics of the waves (Lines 157-179 in the revised version).
- We think the impact of pre-existing TC or pre-TC disturbances on the wave identification would be small simply because their amplitude in vorticity is much smaller than for TCs that have become named storms. In this study, we used the pre-genesis disturbance identified in ERA5 as the ‘pre-TC genesis’ by tracing the observed TC tracks backwards in time using reanalysis data. The weak stage of the TC lifecycle is expected to provide less contamination in the wave identification. However, as the reviewer has mentioned, we cannot entirely rule out the impact of pre-existing and co-existing TCs on waves. This is clarified in Lines 269-273 in the revised version.
- Regarding the Easterly waves. Yang et al (2018) showed that across the equatorial Atlantic Easterly waves are well described by a superposition of R1+R2. The reason is

that the African Easterly Jet only exists over the continent and as African Easterly Waves leave the continent they move into an easterly flow with much less shear and their structure loses the tilts characteristic of AEWs. Vorticity centres can develop within the cyclonic “pouch” of easterly waves and move at a similar speed to R1 and R2, which is consistent with the finding in Yang et al. 2018. We find this behaviour as well in Figure 3. We added another sentence to clarify this point (Lines 161-164 in the revised version).

- We agree with the reviewer that all methods have caveats and uncertainties, including ours.

We hope these improvements and clarifications will address the viewer’s comment.

Specific Comments:

L1: I don’t think it makes a lot of sense to say “Dynamical equatorial waves” because all waves are dynamical, by definition. I think that the authors are trying to make a distinction between “coupled” and “dry” EW, but I think, for the most part ,they could just say equatorial waves. We thank the reviewer for this comment. In the paper, ‘dynamical equatorial waves’ are referred to as ‘wind-derived equatorial waves’ due to the wave identification method. Yes, these equatorial waves could be convectively coupled (wet) or dry waves, because the associated large-scale winds in our method are not necessarily coupled with convection. We agree that, for simplicity, in the title we just call them equatorial waves.

L52-55: This statement needs some clarification/re-wording. If convectively coupled EW were the main precursor for TC, then I agree with your statement. But if “dry” EW are important, then the inadequate simulation of convection should not be any barrier for leveraging EW for extended predictions of TC events.

Thanks for pointing this out. We meant to say that an inadequate simulation of convection could cause errors in convection-dependent EWs but this might not necessarily cause bias in dynamical EWs. We are sorry for the confusion. In the revised version of the manuscript, we removed this sentence and instead added a statement on the necessity of having multiple approaches of EW identification, which was also kindly suggested by the reviewer (please see Lines 46-49).

L122-123: Have you looked at longitude-time diagrams of wind anomalies (without the EW filtering) to see if there is evidence of these westward moving disturbances without filtering for them? Also, a similar composite for TCs that are not associated with the EW identified here could offer a nice contrast to the cases where a EW precursor is identified.

We appreciate the reviewer for this useful suggestion to further improve the novelty of our paper. We carefully addressed this point in the revised version of this manuscript.

- First, we computed wind anomalies before applying wave filtering, as the reviewer suggested. The corresponding Hovmoller diagrams are included in supplementary Figs4-5. We found that EWs diagnosed by our method capture well the unfiltered wind anomalies, especially in the lower troposphere and in the Northern Hemisphere. On the other hand, we also found that compared to the unfiltered wind anomalies, EWs are better related to TCG events in terms of both longevity and significance level. This might be related to a higher level of noise in the unfiltered wind anomalies. We clarified this point in Lines 148-155 in the revised version

- Secondly, we also added the Hovmoller diagrams for TC events that are matched out-of-phase to EWs in supplementary Fig3. It is clear that there are much fewer 'out-of-phase' TCG events than 'in-phase' events, and that the 'out-of-phase' TCG sits in an anticyclonic area of waves, just opposed to that revealed by the 'in-phase' TCG events. We added another clarification of this in Lines 146-148. Thanks for pointing this good point out!

L355 - 361: While the dispersion and vertical structure are not explicitly imposed, they are implicitly constrained through the choice of trapping scale for parabolic cylinder function. For example, the latitudes chosen to investigate WMRG, R1 and R2 are based on the choice of equatorial radius, which itself is related to the vertical structure of the wave, that also imposes the gravity wave speed. By the way, I don't think the value used for the trapping scale is mentioned anywhere, but maybe I missed it.

We thank the reviewer for raising this point. We added clarification on the meridional trapping scale in the Methods section (Lines 453-456). Note that there is only a known link between the trapping scale, vertical structure and zonal phase speed under the conditions met by the basic equatorial wave theory which includes resting atmosphere and uniform static stability. In a realistic situation with wind shear and variation in static stability it would not be possible to predict the vertical structure or phase speed, given the meridional scale. However, we do take the point that they will be related implicitly somehow. We further noted this point in Lines 459-463.

L363: If the projection is applied independently to each variable, does that mean that the WMRG, R1 and R2 are not orthonormal? Either way, a statement clarifying this point might be useful here.

We thank the reviewer for this question, which is a subtle one. The basis functions used for the wave projection, including sinusoidal structure in the zonal direction and parabolic cylinder functions in the meridional direction, are orthogonal and it is this property that enables us to project general data onto the basis functions. This means that the structures labelled as "WMRG", "R1" and "R2" in each variable are orthogonal since they are expressed as a series of the basis functions. However, the waves identified at any instant may not be orthonormal with respect to an energy norm, for example, because the projection coefficients of v , q and r (see Yang et al, 2018 for definitions of these variables) may not be consistent with normal mode structures and therefore not orthonormal in the sense that normal mode solutions are. An even more subtle point is that the "normal modes" obtained from the theory on a resting atmosphere will interact if there is shear in the background flow because they are then themselves non-normal with respect to energy. We have added a clarification in the revised version of the manuscript for this point (Lines 450-453).

Reviewer #2 (Remarks to the Author):

The authors hypothesize that equatorial waves (EWs) enhance the background convection, favouring tropical cyclone (TC) generation. To find out how the two features are related, they combine two methods for identifying EWs and TCs in ERA5 data in 1980-2018.

The EWs are analyzed using the method of Yang et al. that projects winds and geopotential fields on spatial structures of equatorial waves from adiabatic shallow-water equations on the equatorial beta plane. The method is one of methods discussed in a recent review paper by Knippertz et al.

The three westward-propagating waves (R1 and R2 Rossby waves and the WMRG wave) are compared in time and space with the outputs of the TC tracking algorithm. Depending on the relative location of vorticity maxima and winds associated with the three waves, the events are considered “matched in-phase” or “out-of-phase”.

Based on the collocation of EWs and TCs, the authors conclude that westward-moving EWs are responsible for 60-70% of TC genesis events and that EWs have the strongest effect on TC genesis in the North Pacific and North Atlantic. They also argue that EW signals are identifiable up to two weeks ahead and could be used as reliable precursors to TC activity, indicating an unprecedented potential for improving medium-range prediction of TCs.

The large EW variance found by the authors and the EW longevity are questionable given figures 16-17 in Knipperz et al. and their discussion of EW filtering.

We are not entirely clear on what the reviewer means here. It is true that Knippertz et al (2022) show a large difference in EW variance when using different EW identification methods on the same data. However, here we use the same method based on spatial projection onto 2D parabolic cylinder functions (2DS-PCF) as was shown in Knippertz et al to identify high variances associated with equatorial waves. So the results presented here are consistent with Knippertz et al when using the same method and therefore not “questionable” in terms of application of the method. There is of course sensitivity to the method used for equatorial wave identification and this introduces a substantial uncertainty to the analysis. In more detail:

- Knipperz et al. 2022 (Figs 16-17) shows that EWs derived by the 2DS-PCF method can explain higher percentages of unfiltered variance in winds than those from the FWF-PCF method, partially because the former method uses a much broader filter domain. In our paper, the wave identification method is in line with the 2DS-PCF method in Knipperz et al. 2022.
- However, there are some differences in the details of the method used here relative to Knipperz et al. 2022. In our paper, the broad band filter is wavenumber 3-40 and period 2-10 days, while in Knipperz et al. 2022 the 2DS-PCF method uses a filter with wavenumber 1-15 and period 2-30 days. Also, in our paper, the waves are confined to those waves that are concurrent with TCs. In our TC-wave matching process (see the Methods), we apply a threshold (0.5 times standard deviation of V) to wave amplitude when selecting the concurrent waves, ensuring that the waves are both ‘active’ and identifiable early. In other words, here we focus on TC-related waves, which are expected to be climatologically stronger than the overall waves.

In the revised version of the manuscript, we added clarifications in both the Discussion section to address these uncertainties (Lines 266-269, 275-278)

The matching algorithm requires more detailed analysis since the three EW types are largely defined by vorticity.

We appreciate the reviewer pointing this out. Actually, when we optimized the matching algorithm in our analysis for global TC events, we had tried many different combinations of variables, thresholds, and relative position of waves and TCG. For example, we investigated how using relative vorticity in defining the phase relationship for the TCs and waves affects the matching, how sensitive the TC-wave relationship is to the position and length of the green sampling bars in Supplementary Figure 14, how sensitive it is to the criteria of the wave amplitude in the matching (e.g., 0.5 times standard deviation of meridional wind speed). We agree that these sensitivity analyses are fundamentally important for introducing a new method to match the two systems. We should have included (but obviously did not) all these in the first version of the manuscript.

In the revised version, we added two paragraphs in the Methods section (Lines 519-538) and the Discussion section (Lines 262-269) to clarify these points. We hope this will address the viewer's comment.

The main issue with the manuscript is however, that the mechanism by which the two processes, EWs and TC are coupled, is left completely undiscussed. It is unclear how EWs are "responsible for TC genesis events", and how this analysis indicates "an unprecedented potential for improving medium-range prediction of TCs". Given several critical studies of EW forecasting cited in the paper, and availability of ERA5 forecasts, it should be possible to extend the analysis to forecast data and to look into the physical mechanism of how the two features may be coupled.

We thank the reviewer for raising this very good point.

- In the revised version of the manuscript, we have carefully addressed the reviewer's comment by evaluating the environment related to TC activity based on the ERA5 data. We analysed relative humidity at 700hPa, convergence and unfiltered wind anomalies at the lower and upper level, conditional on the wave phase. The results are shown in the modified/new Figures 4 ,6, and in Supplementary Figures 6-8. We concluded that the TC-wave relationships are plausibly associated with the modulation of environmental conditions by the equatorial waves. We added the new analysis in the Results (Lines 181-202), and added a summary in the Conclusion sections (Lines 259-260), and Abstract (Line 19).
- In the revised version, we did not include an evaluation of forecast data to interpret the TC-wave relationship as we are not convinced this will bring much added information over and above that from using the analysing data. From our previous studies, many forecast models (both NWP and climate models) have significant errors and uncertainty in simulating TCs (e.g., Feng et al., 2019, 2020) and equatorial waves (e.g., Yang et al, 2021). Including models in this paper may dilute the research focus and increase the complexity (also length) of the paper. However, we do appreciate the reviewer for this interesting comment, which will inspire our future work.
- With respect to the reviewer's comment: *It is unclear... how this analysis indicates "an unprecedented potential for improving medium-range prediction of TCs"* (this is also pointed out by the 3rd reviewer), we removed the '*unprecedented*' statement as we agree that this is overstated based on the present study.

We hope these improvements will address the reviewer's comment.

Reviewer #3 (Remarks to the Author):

The manuscript reports on the connection between westward-moving dynamical equatorial waves and global tropical cyclone (TC) genesis and intensification, identified using 39 years of ERA5 reanalysis data. The most noteworthy result is that these waves (WMRG, R1, and R2) are all found to be useful precursors for the occurrence and intensification of TCs. Other interesting findings are the heightened relationship for stronger genesis events, and the variations by region. Composites of horizontal wind fields yield further insights into the role of these equatorial waves during and shortly after the time of genesis.

The work is original, of significance to the field, and it maintains a high standard. The approach here of mapping the dynamical fields in the reanalysis data onto the theoretical westward-moving wave solutions is novel. It mostly complements the literature, which generally selects equatorial waves via their convective characteristics (OLR etc.). There are a few contradictions with the state of the field, indicated below. Overall, the work supports the conclusions and claims for WMRG, R1, and R2. The methodology mostly seems sound, with appropriate statistical and sensitivity tests to confirm the robustness of the results. The figures are clear. The manuscript is written transparently for the results to be recreated by an interested reader.

Overall, my recommendation is for acceptance subject to the following revisions:

We appreciate that the reviewer recognizes the research value of our paper. In the revised version of the manuscript, we have considered and carefully addressed the reviewer's comments.

Major Comments:

1. It is reported in the manuscript that “We also evaluated the effect of eastward-moving Kelvin waves on TC activity, and found that Kelvin waves do not have a clear impact on either TC genesis or TC intensification, associated with the fast separation of the TCs and Kelvin waves.” In contrast, papers such as Schreck (2015) demonstrate the linkage between Convectively Coupled Kelvin Waves (CCKWs) and tropical cyclogenesis. Lawton et al. (2022) recently expanded on this, using composites based on 39 years of ERA5 data, to demonstrate the role of CCKWs on African Easterly Wave behavior. These and other papers cited therein are forming a consensus on this relationship. In the manuscript, are the Kelvin Waves convectively coupled, or are they more idealized “dry” Kelvin Waves? To support their claim above, the authors ought to summarize their methodology for finding the Kelvin Waves, and show the results (in the supplement, at least). Otherwise, I recommend that the authors remove their statements on Kelvin Waves if no evidence is provided to back them up.

Thanks for pointing this out. We have read the references mentioned above and recognise the complexity of Kelvin wave association with TCs. Since Kelvin waves propagate eastwards, typically in the opposite direction to the TC movement, we expect the interaction to be more transient. It was certainly less clear in our analysis than the relationship between westward waves and TCs. Thus, we have decided to focus the paper entirely on the westward waves. The

statement related to Kelvin waves has been removed in the revised version, as the reviewer suggested, to avoid misleading.

2. The declaration of “TCG” in ERA5 employed here is inconsistent with how the TC field thinks about genesis. It is certainly beneficial here to include pre-TC disturbances in the ERA5 data, rather than beginning with the Best Track values of position and intensity. However, the time difference (average of 4.6 days) between the observed (IBTrACS) genesis and the “ERA5 TCG” is concerning. This is properly acknowledged in line 314, where the authors state that “ERA5 TCG is the first point of the identified TC track, which is an earlier stage than the genesis in IBTrACS”. This suggests that ERA5 TCG should not be called TCG, and it is not yet part of a “TC track” as a TC is a few days away from forming. A reader might be misled into interpreting the “ERA5 TCG” location as the actual location of the onset of genesis, despite it being an average of 4.6 days, and sometimes even up to 20 days (according to Supplemental Figure 8) prior to the IBTrACS genesis time.

I do see great value in the TC genesis process being utilized. It just does not correspond to the time of actual genesis. How about replacing “ERA5 TCG” with “Pre-Genesis Disturbance” or similar? The terminology is more cumbersome, but more accurate. If this is done, the methodology and results can remain as is, although the explanations will need to be refined to emphasize the processes prior to genesis.

We really appreciate this very good point. We agree that the term “ERA5 TCG” is confusing and could cause misunderstandings, without a clear clarification. In the revised version of the manuscript, in the Introduction, we first introduced the term “pre-TCG” representing ‘Pre-Genesis Disturbance’. We described the vortices at this earlier stage as “pre-TC” features and the first identification in ERA5 as the “pre-TCG” event. Through the paper, “pre-TCG” is used. We added an explicit notice in both the Introduction (Lines 71-76), the Methods (408-416) and Conclusion sections (Line 284) to avoid the terminology confusion. We hope this will address the concern about the term “TCG”.

3. An alternative to Major Comment 2, if the authors feel a strong need to do the phase matching at the genesis point, is to select the genesis location and time that are closest to those in IBTrACS. This would be considerably more work, and perhaps less insightful than the current method which does not use the genesis point for the phase matching.

Yes, using observed TCG in this paper is less insightful. In the revised version, we took the suggestion in Major Comment 2. Please see our response above.

Minor Comments

1. Line 16. What is the “new observational dataset”? This study just uses ERA5, which is no longer new (but still state-of-the-art), and it is not “observational”.

Thanks for pointing this out. We have improved this sentence (Lines 15-16).

2. Lines 25 and 83. While this is an interesting study, it is overstated in places without evidence behind the statements. For example, the last sentence in the Abstract suggests an “unprecedented” potential for improving medium-range predictions of TCs, which seems too simplistic. I suggest removing this sentence, until the potential to add value to medium-range prediction of TCs has been demonstrated.

Thanks. We agree that the statement was made too optimistic based on the present analysis. In the revised version, we have improved these sentences (Lines 23-24, 92-93).

3. Line 43. It seems strange to say that “the identification of equatorial waves is contaminated by the heavy TC-related rainfall”. Complicated, perhaps, but not contaminated. And it is not necessarily restricted to the TC-related rainfall.

Thanks for pointing this out. We have improved this sentence (Lines 43-46).

4. Lines 62 and 63. The TCs are not observed. They are diagnosed in ERA5, as is described correctly in the paragraph beginning on line 302.

Sorry for the confusion. This has been corrected in the revised version (Lines 62-63).

5. While the manuscript is largely well-written and straightforward to follow, there are several minor grammatical and typographical errors.

Thanks. We have gone through the revised version carefully and have tried to correct these errors as much as we can.

References:

Lawton, Q. A., Majumdar, S. J., Dotterer, K., Thorncroft, C., & Schreck, C. J., III., 2022. The Influence of Convectively Coupled Kelvin Waves on African Easterly Waves in a Wave-Following Framework, *Mon. Wea. Rev.*, 150, 2055-2072.

Schreck, C. J., 2015: Kelvin waves and tropical cyclogenesis: A global survey. *Mon. Wea. Rev.*, 143, 3996–4011.

REVIEWERS' COMMENTS

Reviewer #1 (Remarks to the Author):

The authors have addressed my concerns and I have no further comments.

Reviewer #2 (Remarks to the Author):

Thank you for addressing my questions.

I am satisfied with the changes.

Reviewer #3 (Remarks to the Author):

The authors have made substantial revisions to the manuscript, based on the comments of each of the three reviewers. I am largely satisfied with the authors' responses to my comments.

I have a few additional minor comments:

1) Lines 17 and 202: the statement that EWs are "responsible" for a large fraction of genesis events is too strong. This is related to a comment from Reviewer 1 in the first round of reviews, and the issue needs to be addressed. I agree with other statements in the manuscript that EWs are "useful precursors" etc. However, I expect that many of these AEWs would have developed into TCs even if the EWs were not in phase (possibly a delayed genesis, weaker TCs etc.), and hence that it is incorrect to directly attribute the TC genesis events solely to the EWs.

2) Line 103: anticyclonic or cyclonic vorticity?

3) Line 184: I am not sure what "conducted by" means here.

4) Lines 418-421: I do not understand this final sentence, and am not sure it adds any value. It can be removed.

5) Line 488: cyclonic vorticity?

6) The caveats in the ERA5 data need to be explained. For example, I expect that ERA5 estimates of the intensity (surface wind speed) have substantial errors.

7) The revised manuscript is repetitive in places, and can be sharpened.

8) There are still many minor grammatical errors, and confusing or ambiguous sentences. A careful proofread is essential before the manuscript is submitted again.

Responses for Nature Communications: Paper # NCOMMS-22-31072A

Title: Equatorial waves as useful precursors to tropical cyclone occurrence and intensification

We would like to thank the reviewers and editor for their careful reading and constructive comments on the previous version of the manuscript (NCOMMS-22-31072A). We have worked to address all the comments carefully in this revised version. Our responses to the reviewer's comments are highlighted by blue text in the following.

Reviewer #1 (Remarks to the Author):

The authors have addressed my concerns and I have no further comments.

-- We thank the reviewer for reviewing our revised manuscript.

Reviewer #2 (Remarks to the Author):

Thank you for addressing my questions. I am satisfied with the changes.

-- We appreciate that the reviewer confirms this. Thanks.

Reviewer #3 (Remarks to the Author):

The authors have made substantial revisions to the manuscript, based on the comments of each of the three reviewers. I am largely satisfied with the authors' responses to my comments.

-- Many thanks for reviewing our revised manuscript. We are glad to know that the reviewer is largely satisfied with the manuscript.

I have a few additional minor comments:

-- In this new revised version of the manuscript, we have addressed the reviewer's further comments as follows.

1) Lines 17 and 202: the statement that EWs are "responsible" for a large fraction of genesis events is too strong. This is related to a comment from Reviewer 1 in the first round of reviews, and the issue needs to be addressed. I agree with other statements in the manuscript that EWs are "useful precursors" etc. However, I expect that many of these AEWs would have developed into TCs even if the EWs were not in phase (possibly a delayed genesis, weaker TCs etc.), and hence that it is incorrect to directly attribute the TC genesis events solely to the EWs.

-- We thank the reviewer for pointing this out. We have improved our statements in these two sentences (Lines 17 and 262), and also in other places of the manuscript. We agree that it's hard to explain causality of TCG events and EWs in this study.

2) Line 103: anticyclonic or cyclonic vorticity?

-- Thanks for pointing this out. We have replaced this with "negative relative vorticity".

3) Line 184: I am not sure what "conducted by" means here.

-- We are sorry for the confusion. We have replaced this with "associated with" in Line 244.

4) Lines 418-421: I do not understand this final sentence, and am not sure it adds any value. It can be removed.

-- Thanks. We have removed this sentence.

5) Line 488: cyclonic vorticity?

-- Yes. This has been clarified in Line 666 in the revised manuscript.

6) The caveats in the ERA5 data need to be explained. For example, I expect that ERA5 estimates of the intensity (surface wind speed) have substantial errors.

-- We thank the reviewer for this comment. The caveat of ERA5 TCs has been added in Lines 591-595 in the revised manuscript.

7) The revised manuscript is repetitive in places, and can be sharpened.

-- We appreciate the reviewer for this comment. We have gone through the manuscript carefully and improved the readability, including shortening text where, we think, necessary.

8) There are still many minor grammatical errors, and confusing or ambiguous sentences. A careful proofread is essential before the manuscript is submitted again.

-- Thanks. We have read the manuscript again and tried our best to correct these errors.